# GIS-Based Simulation for Landfill Site Selection in Mekong Delta: A Specific Application in Ben Tre Province

Dinh-Thanh Nguyen [1,2] , Minh-Hoang Truong [3,*], Thi-Phuong-Uyen Ngo [2,4], Anh-Minh Le [1,2] and Yuya Yamato [5]

1  Department of Hydrogeology and Engineering Geology, Faculty of Geology, University of Science, 227-Nguyen Van Cu Street, Ward 4, District 5, Ho Chi Minh City 700000, Vietnam
2  Vietnam National University Ho Chi Minh City, Linh Trung Ward, Thu Duc District, Ho Chi Minh City 700000, Vietnam
3  Faculty of Civil Engineering, School of Engineering and Technology, Van Lang University, 69/68 Dang Thuy Tram Street, Ward 13, Binh Thanh District, Ho Chi Minh City 700000, Vietnam
4  Department of Fundamental Geology, Faculty of Geology, University of Science, 227-Nguyen Van Cu Street, Ward 4, District 5, Ho Chi Minh City 700000, Vietnam
5  National Institute of Technology, Fukui College, Sabae 916-8507, Japan
*  Correspondence: hoang.tm@vlu.edu.vn

**Abstract:** The aim of this research is to develop a GIS-based simulation for selecting the most suitable site of solid waste landfill which could help to minimize harmful impacts to the environment and society in the extreme sensitive and complex delta by an integration of geographic information system (GIS) and analysis hierarchy process (AHP) and nine criteria (distance from surface water; depth of ground water table; distance from residential area, land use, distance from main roads, geo-environmental and geotechnical characteristics, distance from historical and tourism sites, and distance from industrial zones). Different from most of the previous studies on the landfill site selection, geology-related criteria including soil types/lithology, soil permeability, and soil depth/soil thickness (soil-structure), which are called geo-environmental and geotechnical characteristics in this research, will be carefully considered, integrated, and evaluated. The AHP was employed to determine the weight of each criterion based on pair weight comparison and its matrix, while a land suitability index (LSI) score was calculated to determine the most suitable site. Moreover, the suitability map was also created which indicated very advantageous, advantageous, rather advantageous, and disadvantageous areas in the study area for landfill siting. Finally, the developed model could be used for supporting planners, managers, policy makers, and local government to make decisions on suitable and effective planning strategies for landfill site selection and could be applied anywhere and especially in other deltas around the world.

**Keywords:** analytic hierarchy process; location choice; relative weight; model builder; planning support

## 1. Introduction

The same as most developing countries in the world, the solid waste (generated by inhabitants) process in Vietnam is still conducted by the traditional method which buries waste at landfills. Because of rapid population increase and urbanization, the amount of waste is more increased, which result in overloaded situations at many landfills in many cities and provinces of Vietnam. Moreover, stench from the landfills is also another serious issue that needs to be solved strictly. In particular, the problem may be more serious if the landfill is placed in the Mekong delta (hereafter, MD), which is a complex and sensitive region in Vietnam. The MD has rather flat topography with altitudes of $+/-2$ m in the average present sea level (hereafter, a.p.s.l), which was divided into two broad parts: the upper part is strongly influenced by flood from the Mekong River system while the lower part is strongly influenced by tidal regimes from the East Vietnam Sea. In addition, the ground water level is so high, which is about 0.0 m in the

a.p.s.l. Therefore, the MD is in dynamical hydro-graphic and -geology, and this is a good condition for spreading contamination. If a landfill is not effectively and suitably sited, contamination can spread everywhere in the MD. Moreover, the landforms, sedimentary structures, thickness and material composition, and geo-environmental and geotechnical characteristics of sedimentary facies on the ground surface, which were formed in the completed marine regression and the flood, have undergone complex changes [1–4]. The above problems indicate that the current planning for landfill sites was not effective due to quick population growth and economic development in big cities in Vietnam as well as provinces in the MD. Furthermore, locating a landfill in MD should be carefully considered and estimated based on geo-environmental and geotechnical characteristics.

This research, therefore, developed a GIS-based model for landfill site selection based on estimating nine criteria including sub-criteria using the analytic hierarchy process (AHP) method. The model may become a powerful tool to solve the problem of landfill site selection and help planners, managers, and local government proposing suitable and effective planning strategies for landfill construction. Moreover, using the model in the first stage of landfill projects, which names investigation and location choice, will expedite tasks and result in reduced costs.

To date, there are many studies that have been conducted on the site selection of landfill around the world in the literature. Most of them applied the integration of geographic information system (GIS) and analytic hierarchy process (AHP) as the research methodology [5–35]. While some others also used GIS with different additional methods, such as integration of weighted linear combination (WLC) and AHP [36,37]; AHP and fuzzy logic technic [38–41]; AHP and simple additive weighting (SAW) [42]; AHP and fuzzy TOPSIS [43]; decision-making trial and evaluation, and analytical network process [44]; AHP, SAW, and combinative distance-based assessment [45]; LEFT and AHP [46]; AHP, weighted linear combination, and ordered weighted averaging [47]; AHP, SAW, and straight rank sum [48]; AHP and ratio scale weighting [49]; analytic network process, fuzzy logic, and ordered weighted averaging [50]; HFLTS-based (hesitant fuzzy linguistic term sets) TOPSIS [51]; fuzzy AHP and fuzzy TOPSIS [52]; fuzzy logic spatial modelling [53]; and GIS and MCDA [54,55].

These verified that the integration of GIS and AHP is the most common approach for identifying suitable sites for landfill in specific and other types of facilities in general, such as deep-water port [56], solar power farm [57], and wind farm [58]. In addition, to enhance public participation for land suitability evaluation, several studies applied the integration of GIS and AHP in the World Wide Web environment, which was also known as Web-based multi-criteria spatial decision support system [59,60] or Web GIS-based multicriteria decision analysis [61]. With these applications, users could select and adjust parameters (criteria) according to their preferences and knowledge, which could result in generating different scenarios.

Because of impacts on environment, society, human health, as well as construction cost, landfill siting requires consideration and evaluation of many criteria to identify the most suitable site. From the 52 referred papers from literature, the most used criteria include distance from roads (51 per 52 papers, hereafter 51), distance from surface water (49), distance from residential areas (49), slope (46), land use (44), ground water (42), geology (40), distance from historical sites (26), distance from environmentally protected areas (24), distance from faults (23), and distance from airports (22). Moreover, Ozkan et al. [62] also conducted a review on 106 studies of landfill siting and showed the same results on most common used criteria which were surface water/wet land, distance from urban areas, distance from road, slope, land use, geology, ground water, distance from fault zone, distance from airport/heliport, and distance from environmentally protected areas. Among them, the geology related criterion (hereafter, geology criterion), which were referred to as soil types or lithology, soil permeability, and soil depth, was considered as a quite important criterion for landfill site selection. Several previous studies also evaluated the geology criterion as the top three important criteria, such as [8,12,15,20,22,24,27,37,44,45,63]. However,

from the literature, most of the studies only considered a single factor of soil type/lithology while few authors considered a couple of soil type/lithology with soil permeability or with soil depth/soil thickness [15,17,20,22,44,46]. For the former, because all the study areas were not delta areas, authors only based on the soil types to identify the soil permeability which was important factor for landfill siting, such as clay having low permeability and sand having high permeability. However, from the geo-environment and geotechnic aspects, considering the single factor of soil type/lithology was insufficient to apply to delta areas, especially in MD because of covering Holocene sediments, which are recently formed sediments with very soft surface soils. Although the Holocene sediments include sedimentary faces with surface clayey and silty soil layers (impermeable layers), they are in different consolidation levels, which result in different values of permeability [1,64]. Moreover, each sedimentary face was formed in a different specific environment and owned typical geotechnical properties [64]. The consolidation levels of each sedimentary face were dependent on the formed time [1]. In addition, the thickness of surface soil layers also affects the soil permeability which could cause pollution. A quite permeable and thick soil layer, for instance, may have higher potential for landfill siting than an impermeable and thin soil layer. Therefore, the soil type could not reflect geo-environmental and geotechnical characteristics, which play an important role for landfill siting. In the latter, although few authors considered thickness [44,46] or permeability [15,17,20,22] of soil layers combined with soil type, analysis, and evaluation were not conducted in detail. For the soil thickness, the authors only mentioned that the thicker the soil layer is, the higher the score is [44,46], and this did not present any certain value of soil thickness in the case studies. Some authors evaluated soil permeability based on the soil types [15,17] while others showed maps of soil permeability with certain values of permeability [20,22]. However, as mentioned above, the soil thickness also affects the soil permeability in some cases.

From the above explanation, the three factors: soil type/lithology, soil thickness, and soil permeability have a reciprocal relationship and should be combined and evaluated together for landfill siting, especially in the case study of MD. This research, therefore, proposed a new criterion, namely geo-environmental and geotechnical characteristics by integrating the three above-mentioned factors of geology (soil type, soil thickness, and soil permeability), which is also considered as a new contribution to the literature. The data for this criterion were obtained from in situ (the piezo-cone penetrometer—CPTU) and lab test (one-dimensional consolidation, incremental loading oedometer test) which could provide necessary and precise information on the conditions of environmental geology, geotechnic, and sedimentary geology in the study area. Furthermore, as usual, for a landfill construction project, the landfill siting is considered as the first stage of the project, which is followed by the second stage of the field investigation on geo-environment engineering, geotechnic engineering, and sedimentology. In the case in which the first stage did not evaluate sufficiently geo-environmental and geotechnical characteristics, it may cause many difficulties, such as soil treatments and improvements needed at the selected site before construction, or in some cases, the selected site must be changed and all the works must be conducted again, which make construction expenditure and time significantly increased compared with the initial investment plan. Therefore, considering and evaluating the criterion of geo-environmental and geotechnical characteristics in the first stage of site selection would make landfill siting more appropriated and effective, and minimize risks during the landfill project construction, as well as save expenditure and time.

In addition, because the landfill construction cost is very expensive, the requirement of the area of selected site should be large enough to meet the population increase and development plan from the local government in the future. Therefore, based on plans for industrial, residential settlement, and tourism development, and population increase in the study area of Thanh Phu district, Ben Tre province, the requirement of the area of selected sites in this research was more than 10 ha, which corresponds to medium-size land fill and could serve a population of 100,000 to 500,000 people (according to Vietnamese construction standard for solid waste landfill: TCXDVN: 216-2001, hereafter TCXDVN) [65].

This paper is laid out as follows: the method of developing a model is presented in Section 2, followed by results and discussion in Section 3, and conclusions are made in Section 4.

## 2. Methodology

### 2.1. Study Area

Thanh Phu district is one of three coastal districts of Ben Tre province to which the Mekong River Delta region of Vietnam belongs, and is influenced strongly by operations of the Ham Luong river system, the Co Chien river system, and operation of the tide of the East Vietnam Sea (Figure 1), and has several surface sedimentary facies. Therefore, Thanh Phu district has enough factors and complex conditions as a case study for establishing the GIS-based simulation for the optimal landfill site selection in the MD. The district has an area of 411 km$^2$ and a population of 127,800 people (in the year of 2019). According to the development plan in the future from the local government, the district will focus on industrial zones, tourism, and new residential settlement development which will attract more labor sources, tourists, as well as residents from other districts or provinces. Furthermore, combining with natural population increase will result in a big population in the future.

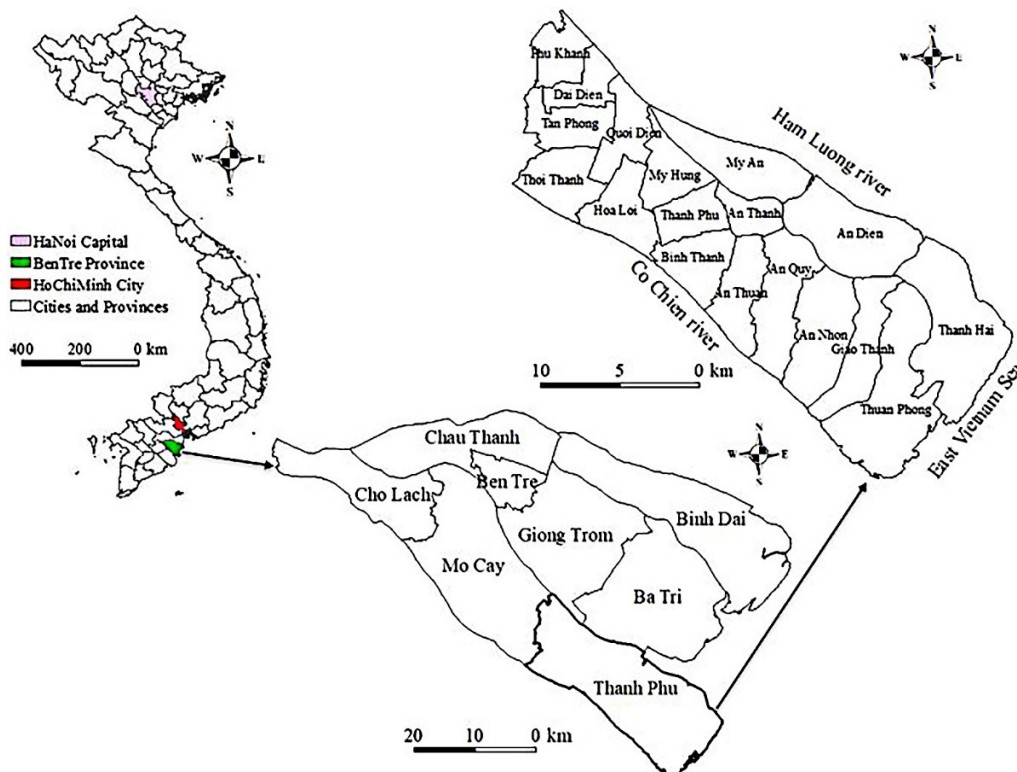

**Figure 1.** Geographic location of Ben Tre province (**left**), its districts (**middle**), and Thanh Phu district's communes (created using WGS_1984_UTM_Zone_48N).

Now, there is also an existing landfill with an area of 12.2 hectares in Thanh Phu townlet, Thanh Phu district where 25 to 30 tons of garbage arrive each day. The landfill is in the overloaded status and is being improved by the local government. Moreover, problems on pollution are also recorded from this landfill site. Therefore, from the above explanation on the development plan, population increase and the issue of existing landfills, it is very necessary and important to propose landfill site planning in Thanh Phu district for future sustainable development.

However, due to belonging to the MD region, the district has a dense surface water network (rivers, canals, streams) with a density of 2.5 km/km$^2$ and a low topography with

a height ranging from −2 m to 2 m in the a.p.s.l. Moreover, soft soils, which are clay or silt with unconfined compression strength, $q_u = 0$–50 kN/m$^2$ [66], are also a characteristic of the region, and the residents live along the road network. These will become challenges for landfill site selection which will be solved in this research.

## 2.2. Simulation Model Design

A procedure for simulating landfill site selection is presented in Figure 2, which is conducted in the following steps. Firstly, eight criteria, including distance from surface water, depth of groundwater table, distance from residential areas, land use, geo-environmental and geotechnical characteristics, distance from main roads, distance from history and tourism sites, and distance from industrial zones, are selected and considered by referring to 52 existing studies and the current situation of the district. These criteria, then, were divided into 30 sub-criteria, whose weights will all be determined by the AHP method. Moreover, the ArcGIS software is also employed for processing the input data, creating criteria layers as well as calculating land suitability index (LSI) scores by using "model builder" function. Finally, the site with the highest LSI score among the sites with an area greater than 10 hectares was chosen as the most suitable landfill site.

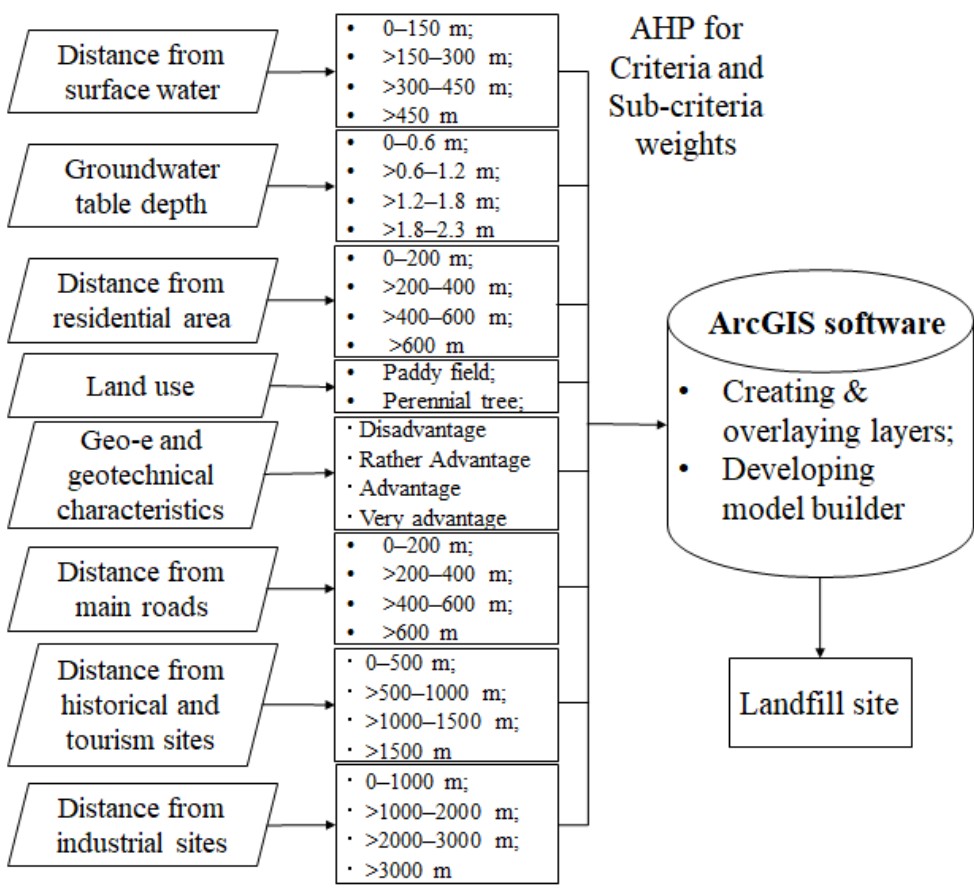

**Figure 2.** Procedure for simulating landfill site selection.

## 2.3. Data Collection and Processing

To create data for the selected criteria, essential data were collected, such as land use map, boreholes, geomorphology map, and road network (Table 1). Apart from "road network data", which are original "shapefile" data, other data were processed and converted into "shapefile" data in ArcGIS software before inputting into the developed model. For the land use map with an original AutoCAD file type, after inputting into ArcGIS software, each sub-layer of the map corresponding to each type of land use was extracted based on attribute table, and then saved as "shape file" data. The data of surface water; residential

areas; history and tourism sites; and industrial zone sites were also created in the same way. The collected boreholes, which represented information on investigated sites with coordinates, depth of groundwater table, and thickness of surface soil layers, were inputted in ArcGIS using their coordinates. All of the borehole information was also manually added as borehole attributes. Then, the maps of depth of groundwater table and thickness of surface soil layers were created by using the IDW interpolation. Finally, the geomorphology map was digitized from the "JPEG file" into "shape file" data.

**Table 1.** Collected data.

| No. | Name of Data | File Type | Source |
| --- | --- | --- | --- |
| 1 | Network of surface water | Shape file | Extracted from land use map |
| 2 | Depth of groundwater table | Shape file | Created from bore-hold data |
| 3 | Distribution of residential areas | Shape file | Extracted from land use map |
| 4 | Land use map | AutoCAD file | Department of Natural Resources and Environment, Thanh Phu district, Ben Tre province |
| 5 | Road network | Shapefile | https://www.geofabrik.de (accessed on 15 March 2021) |
| 6 | Geomorphological map | JPEG | [67] |
| 7 | Bore-hole | Excel file | Department of Hydrogeology and Engineering Geology, University of Science, HCM-VNU |
| 8 | History and tourism sites | Shape file | Extracted from land use map |
| 9 | Industrial zones sites | Shape file | Extracted from land use map |

*2.4. Criteria Description*

This research considered eight criteria for the suitable land fill siting, which includes distance from surface water, depth of ground water table, distance from residential area, land use, main roads, geo-environmental and geotechnical characteristics, distance from historical and tourism sites, and distance from industrial zones. These criteria were selected, evaluated, and set in order of importance levels based on the literature. Moreover, due to difference of the study area, the research also proposed the value of sub-criteria which are different from those in the literature. The criteria on slope and elevation, which are considered as one of the most important criteria and suggested in many previous studies, were not considered in this research because these criteria values were less than 1° and from −2 to 2 m height, respectively, which were evaluated as advantages for landfill site selection. The detailed evaluation of the criteria was represented in the sections below.

2.4.1. Distance from Surface Water

The landfill site should not be near the surface water network, such as rivers, channels, canals, and ponds because leachate from the landfill will result in water contamination and affect people's health. Depending on different study cases, this criterion was considered as the most important or quite important or less important one compared to others. From the referred literature, there were 9, 20, and 30 studies, which evaluated distance from surface water as the top 1, 2, 3 criteria for landfill siting, respectively. Moreover, Thanh Phu district, located in the Mekong River delta, has a dense river-channel-canal network (Figure 3a). Therefore, surface water contamination is considered as a major concern in landfill site selection in this study. From the literature, the restriction buffer between the landfill and surface water network widely varied case by case, such as 100 m [37], 150 m [11], 200 m [43], 250 m [7], 300 m [5], 400 m [14], 500 m [38], 1000 m [10], and 2000 m [26]. According to TCXDVN, although there was no mention of a certain restriction distance between them, it is said that the landfills should be as far as possible from surface water networks. Therefore, in this study, the buffer of 150 m from the surface water network was considered as a restriction zone which preferred disadvantageous areas for landfill siting [11,20,47]. The other three distance categories, which included >150–300 m, >300–450 m, and >450 m, were also created to indicate rather advantageous, advantageous, and very advantageous areas for landfill site selection, respectively (Figure 4a).

### 2.4.2. Depth of Groundwater Table

Similar to surface water networks, the depth of groundwater table is an important criterion because it is also related to leachate from landfill, which could infiltrate into the ground and cause groundwater contamination. The deeper the groundwater table is, the lesser the opportunity for groundwater contamination is. Based on 52 collected boreholes (Figure 3b), the depth of groundwater table in the study area arranged from 0.3 m to 2.3 m. This study, then, created four categories of groundwater table depth which were 0–0.6 m; >0.6–1.2 m; >1.2–1.8 m; and >1.8–2.3 m (Figure 4b) to indicate disadvantageous, rather advantageous, advantageous, and very advantageous areas, respectively.

### 2.4.3. Distance from Residential Areas

The proximity between landfill site and residential areas will cause negative impact on not only the environment, such as odor and noise pollution, which affect residents' health and living activities, but also the landscape. The further the distance is, the lesser the impact is. From the literature, the minimum distances from the residential area were different from different study areas, such as 100 m [38]; 200 m [43]; 300 m [47]; 500 m [45]; 1000 m [50]; 1500 m [55]; 2000 m [23]; 2500 m [44]; 3000 m [33]. According to TCXDVN, residential areas were divided into urban area, mountainous residential area, and residential area of alluvial plain and midland. The two former ones were set as 3 km for the small and medium landfill sites, while the latter was not assigned. Moreover, Thanh Phu is a rural district which is located in the alluvial plain, and residents live along the road network (Figure 3c). Therefore, in this study, a distance less than or equal 200 m from the residential areas is reported as a disadvantageous area while other distance categories, >200–400 m; >400–600 m; >600 m [25,43] (Figure 4c) are considered as rather advantageous, advantageous, and very advantageous areas, respectively.

### 2.4.4. Land Use

Landfills should be placed on suitable land use types, which were assigned by law. Unused land [5,21,42], barren land [6,25,39], vacant land [31,43], wasteland [17], and pasture [30] are priority areas for landfill siting, which are often mentioned in literature. Moreover, in some cases, the agriculture land was considered as moderately suitable [16,28,43]. In this study, based on the land use map collected from the local government (Figure 3d), suitable land use types included paddy field and perennial tree land, whereas other land types were considered unsuitable or prohibited areas, such as residential area, wasteland (the existing landfill), aqua-culture area, cemetery land, protected forest area, productive forest area, and land for national defence and security purposes. Therefore, the perennial land was evaluated as an advantageous area while the paddy field corresponded to a rather advantageous area (Figure 4d).

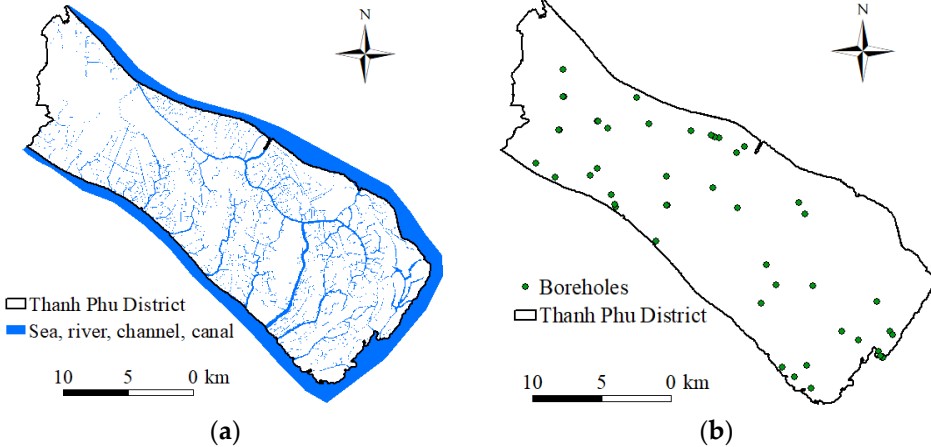

**Figure 3.** *Cont.*

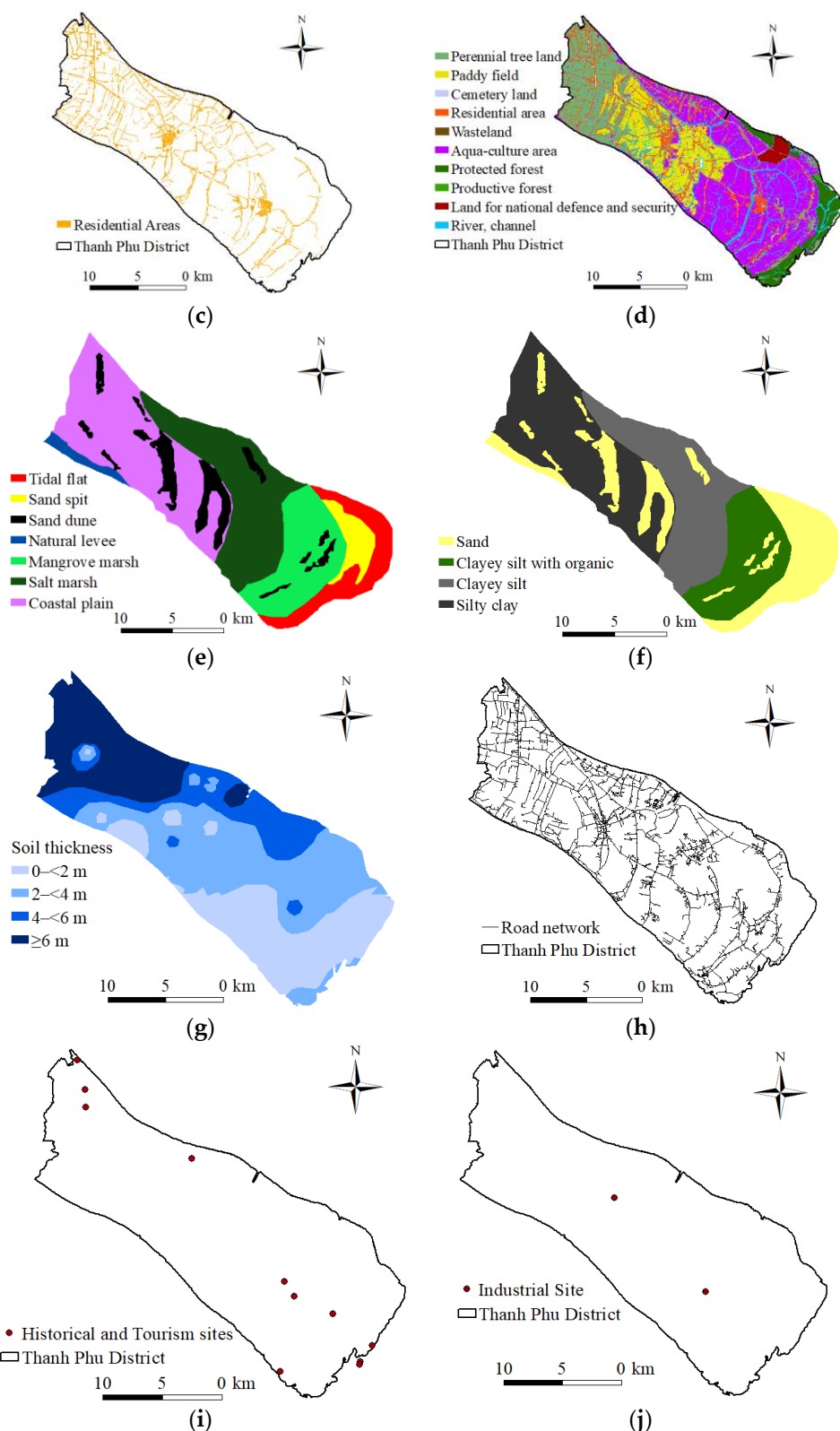

**Figure 3.** Collected data for describing criteria: Surface water network (**a**); locations of collected boreholes (**b**); distribution of residential areas (**c**); land use types (**d**); geomorphological maps (**e**); surface soil types based on geomorphological map (**f**); distribution of surface soil thickness (**g**); road network (**h**); locations of historical and tourism sites (**i**); locations of industrial sites (**j**).

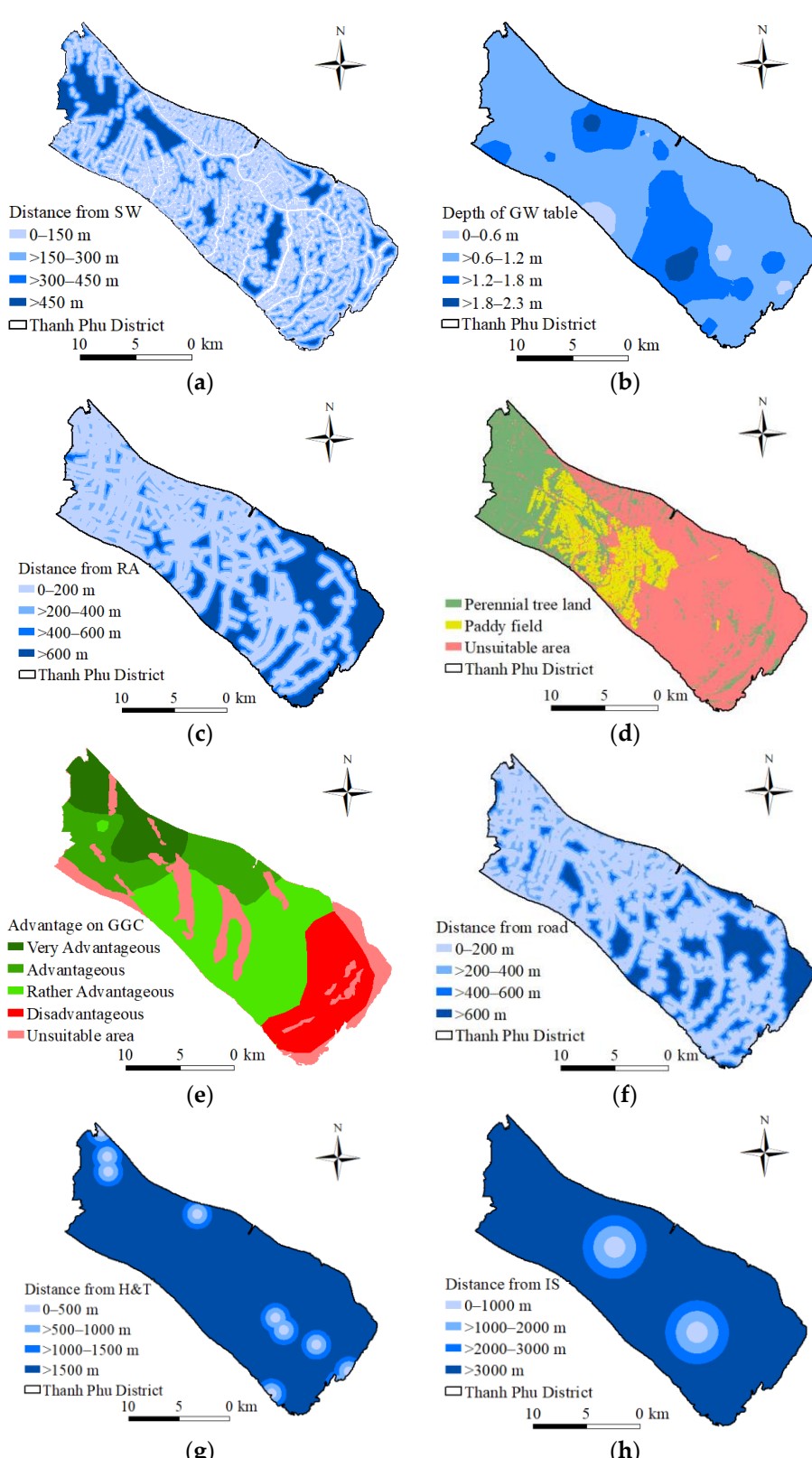

**Figure 4.** Considered criteria in the model: distance categories from surface water (**a**); distribution of ground water level (**b**); distance categories from residential areas (**c**); suitable land use types (**d**); advantages for geo-environmental and geotechnical characteristics (**e**); distance categories from main roads (**f**); distance categories from historical and tourism sites (**g**); distance categories from industrial sites (**h**).

### 2.4.5. Geo-Environmental-Geotechnical Characteristics

In this research, geology criteria in the literature, including soil types, soil thickness, and soil permeability, which was called geo-environmental-geotechnical characteristics, were integrated, and evaluated for landfill siting in Thanh Phu district. Firstly, the collected geomorphological map [67], which showed seven geomorphological units in the study area (Figure 3e) including natural levee, coastal plain, sand dune, salt marsh, mangrove marsh, sand spit, and tidal flat, was used for identifying surface soil types of sand—silty sand, soft medium-high plasticity silty clay with few fine sands (hereafter silty clay), sand, soft medium plasticity clayey silt (hereafter clayey silt), soft medium plasticity clayey silt with organic (hereafter clayey silt with organic), sand, and mud–sand respectively (Figure 3f). Among these soil types, silty clay and clayey silt covered most of the study area and were considered as suitable places while sand was reported as a restriction for landfill siting. Organic clayey silt, belonging to the mangrove marsh, is recently formed sediment and is very soft and unconsolidated soil. With respect to geotechnical characteristics, this soil type is estimated as a disadvantageous place for landfill siting.

Secondly, the two soil types—silty clay and clayey silt—have different material components, different formation processes, and different consolidation levels that make the soil permeability different [1,64]. According to the results on permeability tests, the permeability values of silty clay in the coastal plain are $994.49 \times 10^{-9}$ cm/s to $1231.82 \times 10^{-9}$ cm/s, and clayey silt in salt marsh are $994.49 \times 10^{-7}$ cm/s to $1231.82 \times 10^{-8}$ cm/s, respectively.

Thirdly, apart from soil types and soil permeability, the thickness of surface soil layers also plays an important role for landfill siting, which was identified based on the collected boreholes and are arranged from 0 to more than 6 m (Figure 3g) in this research. For instance, the silty clay layer with lower permeability is thin, while the clayey silt layer with higher permeability is thick. Therefore, this research set a coefficient t based on the permeability of silty clay and clayey silt as 1.5 and 1, respectively, and created four categories of soil thickness as 0–<2 m, 2–<4 m, 4–<6 m, ≥6 m with their coefficient *n* of 1, 2, 3, 4, respectively. The final score of advantages on geo-environmental and geotechnical characteristics ($S_{GGC}$) (Table 2) is calculated as the following equation:

$$S_{GGC} = t * m \tag{1}$$

t: coefficient based on permeability; m: coefficient based on soil thickness.

The $S_{GGC}$ will be calculated at each borehole; an IDW interpolation tool, then, was applied to present a distribution of advantages for geo-environmental and geotechnical characteristics in the study area (Figure 4e).

**Table 2.** Suitability on geo-environmental and geotechnical characteristics.

| Score | Advantage for Geo-Environmental and Geotechnical Characteristics |
|---|---|
| <2 | Rather advantageous |
| ≥2 and <4 | Advantageous |
| ≥4 | Very Advantageous |

### 2.4.6. Distance from Main Roads

Proximity from main roads will not only lead to cost saving for new road construction [39] and waste transportation to landfill sites, but also mitigate environmental issues, such as odor and noise pollution. Therefore, the closer the distance from the main road, the higher the potential for developing landfills. According to TCXDVN, the distance from main roads should be over 100 m, which is applied for small and medium landfills, while those of big and very big landfills are >300 m and >500 m, respectively. Moreover, the literature showed various ranges of restricted distance between main roads and landfill sites, which included 80 m [9], 100 m [10], 300 m [5,38], and 500 m [8,42]. Furthermore, as it was mentioned in Section 2.4.3 that residents live along the road network, this study created four distance categories from the main roads (Figure 3h) that were 0–200 m, >200–400 m, >400–600 m, and

>600 m (Figure 4h) corresponding to disadvantageous, very advantageous, advantageous, and rather advantageous areas, respectively.

2.4.7. Distance from History and Tourism Sites (H&T Sites)

Sitting landfills near the historical and tourism sites seriously affects the surrounding landscape from these sites, tourism activities, as well as attracting tourists due to odor and noise pollution. Although the TCXDVN did not mention the restriction distance from these sites, most literature pointed out that a landfill should be placed over 500 m away from the historical and tourism sites (Figure 3i) [8,10,11,14,22,26,27,38], and this restricted distance is far more suitable. Therefore, four distance categories were created in this study, which are less than or equal to 500 m, >500–1000 m, >1000–1500 m, and >1500 m (Figure 4g), corresponding to disadvantageous, rather advantageous, advantageous, and very advantageous areas, respectively.

2.4.8. Distance from Industrial Zone Sites

Industrial zones are places where many people concentrate and work together. Therefore, landfills should be sited at a suitable distance from the existing industrial zones (Figure 3j) in order to dismiss its impacts on people's health due to odor and dust. The further the distance, the higher the potential for developing landfills. According to TCXDVN, landfills at a distance of less than 1000 m from industrial zones are restricted. The four distance categories of 0–1000 m, >1000–2000 m, >2000–3000 m, and >3000 m [6,15] (Figure 4h) were created in this study, which corresponded to disadvantageous, rather advantageous, advantageous, and very advantageous areas, respectively.

*2.5. Analytic Hierarchy Process*

The analytic hierarchy process (AHP) is considered as one of the multi-criteria decision-making methods for determining the weight of the selected criteria proposed by Saaty [68]. Specifically, higher weights indicate a more important criterion. Pairwise comparisons of criteria and matrixes of these pairwise comparisons are specific characteristics of this method. Moreover, a fundamental scale of the AHP, with values from 1 to 9 (Table 3), was employed for judging the pairwise comparison.

**Table 3.** AHP scale for pairwise comparison.

| Value of $p_{ij}$ | Explanation |
| :---: | :---: |
| 1 | i is equally important to j |
| 3 | *i* is slightly more important than *j* |
| 5 | *i* is strongly more important than *j* |
| 7 | *i* is very strongly more important than *j* |
| 9 | *i* is extremely more important than *j* |
| 2, 4, 6, 8 | Intermediate values |

After selecting the criteria and conducting the pairwise comparisons, a matrix *A*, then, was created based on the results of these comparisons, which is shown in Equation (2):

$$A = \begin{bmatrix} a_{11} & \cdots & a_{1n} \\ \vdots & \ddots & \vdots \\ a_{n1} & \cdots & a_{nn} \end{bmatrix} \quad (2)$$

$a_{11}$, $a_{1n}$, $a_{n1}$, $a_{nn}$: values of the AHP scale, which are from 1 to 9.
1, ... , *n* are criteria.

Next, a matrix $A_w$ was established based on matrix $A$ by dividing each value of each column by the sum of the values of each column, which is presented in Equation (3):

$$A_w = \begin{bmatrix} \frac{a_{11}}{\sum Col1} & \cdots & \frac{a_{1n}}{\sum Coln} \\ \vdots & \ddots & \vdots \\ \frac{a_{n1}}{\sum Col1} & \cdots & \frac{a_{nn}}{\sum Coln} \end{bmatrix} \tag{3}$$

where $\sum Col1; \ldots ; \sum Coln$ are the sum of values of column 1, ..., column $n$.

Finally, the relative weight of each criterion was obtained by averaging the values of each row in matrix $A_w$, which is represented as matrix $C$ in Equation (4):

$$C = \begin{bmatrix} c_1 \\ \vdots \\ c_n \end{bmatrix} = \begin{bmatrix} \frac{\frac{a_{11}}{\sum Col1} + \cdots + \frac{a_{1n}}{\sum Coln}}{n} \\ \vdots \\ \frac{\frac{a_{n1}}{\sum Col1} + \cdots + \frac{a_{nn}}{\sum Coln}}{n} \end{bmatrix} \tag{4}$$

The values of $c_1, \ldots, c_n$ are relative weights of selected criteria in this research. However, to estimate whether the above weight values were reasonable, the consistency ratio (CR) was calculated using the following steps:

Firstly, the eigenvector with values of $x_1, \ldots, x_n$ was calculated by multiplying matrixes $A$ and $C$:

$$A \times C = \begin{bmatrix} a_{11} & \cdots & a_{1n} \\ \vdots & \ddots & \vdots \\ a_{n1} & \cdots & a_{nn} \end{bmatrix} \times \begin{bmatrix} c_1 \\ \vdots \\ c_n \end{bmatrix} = \begin{bmatrix} x_1 \\ \vdots \\ x_n \end{bmatrix} \tag{5}$$

Then, the maximum eigenvalue was calculated using the following equation:

$$\lambda_{max} = \frac{1}{n} \sum_{i=1}^{n} \frac{x_i}{c_i} \tag{6}$$

$\lambda_{max}$: is the eigenvalue of the pairwise comparison matrix;
$n$: is the number of criteria.

Finally, the consistency index (CI) and CR were calculated as follows:

$$CI = \frac{\lambda_{max} - n}{n - 1} \tag{7}$$

$$CR = \frac{CI}{RI} \tag{8}$$

RI: random CI results, for which the value depends on the number of criteria, are shown in Table 4 below. The RI values were derived from Saaty [68].

**Table 4.** RI values (Saaty 1980) [67].

| $n$ | 1 | 2 | 3 | 4 | 5 | 6 | 7 | 8 | 9 | 10 |
|-----|------|------|------|------|------|------|------|------|------|------|
| RI | 0.00 | 0.00 | 0.58 | 0.90 | 1.12 | 1.24 | 1.32 | 1.41 | 1.45 | 1.49 |

If the CR value is less than or equal to 0.1, the results of the pairwise comparison are reasonable; if the CR value is over 0.1, the pairwise comparison must be revised.

As a result, these criteria were overlaid, and the overall score of LSI was determined by following Equation (9):

$$LSI = \sum W_i \times W_{ij} \tag{9}$$

$W_i$: weight value of criterion $i$;
$W_{ij}$: weight value of sub-criterion $j$ of criterion $i$.

## 3. Results and Discussion

The calculated results of CR showed values of 0.062 and 0.033 for the criteria and sub-criteria, respectively, which proved that the results of pairwise comparisons were reasonable and acceptable. The weights of the criteria and sub-criteria are represented in Table 5.

**Table 5.** Weights of criteria and sub-criteria.

| Criteria | Weight | CR | Sub-Criteria | Weight | CR |
|---|---|---|---|---|---|
| Distance from surface water | 0.325 | | >450 m<br>>300–450 m<br>>150–300 m<br>0–150 m | 0.602<br>0.243<br>0.105<br>0.050 | 0.033 |
| Depth of groundwater table | 0.224 | | >1.8–2.3 m<br>>1.2–1.8 m<br>>0.6–1.2 m<br>0–0.6 m | 0.602<br>0.243<br>0.105<br>0.050 | 0.033 |
| Distance from residential areas | 0.152 | | >600 m<br>>400–600 m<br>>200–400 m<br>0–200 m | 0.602<br>0.243<br>0.105<br>0.050 | 0.033 |
| Land use | 0.101 | 0.062 | Perennial tree land<br>Paddy field | 0.550<br>0.540 | - |
| Geo-environmental and geotechnical characteristics | 0.064 | | Very advantage<br>Advantage<br>Rather advantage<br>Disadvantage | 0.602<br>0.243<br>0.105<br>0.050 | 0.033 |
| Distance from main roads | 0.064 | | 0–200 m<br>>200–400 m<br>>400–600 m<br>>600 m | 0.602<br>0.243<br>0.105<br>0.050 | 0.033 |
| Distance from history and tourism sites | 0.041 | | >1500 m<br>>1000–1500 m<br>>500–1000 m<br>0–500 m | 0.602<br>0.243<br>0.105<br>0.050 | 0.033 |
| Distance from industrial zone sites | 0.028 | | >3000 m<br>>2000–3000 m<br>>1000–2000 m<br>0–1000 m | 0.602<br>0.243<br>0.105<br>0.050 | 0.033 |

As mentioned above on the required area of selected landfill location, in this research, the location with the highest LSI score among locations with an area greater than 10 ha will be chosen for the landfill site. According to the simulated results, the LSI score ranged from 0.109 to 0.488 (Figure 5) in the study area, and the most suitable site had an area of 14.51 ha and LSI score of 0.477 (Figure 6). The site is in My Hung Commune, and it is 3 km away from the existing landfill located in Thanh Phu townlet as well as about 3 km away from the central area of Thanh Phu townlet, which is also a central part of the Thanh Phu district. The detailed criteria of this site included six very advantageous criteria: distance from surface water of over 450 m; >1.8−2.3 m deep from groundwater table; very advantageous geo-environment and geotechnical characteristics; distance from road of 0–200 m; distance from historical and tourism sites of over 1500 m; distance from industrial zone sites of over 3000 m, one rather advantageous criterion: belonging to paddy field, and one disadvantageous criterion: distance from residential area of 0–200 m. As mentioned in the above section of "distance from residential area", Thanh Phu is the rural district and residents are living along the road network. Therefore, although the selected

site located in the disadvantageous distance category of 0–200 m from the residential area, few households are living in this distance category. Moreover, because of living in the rural area, each household owns a big land slot with a big garden around the house.

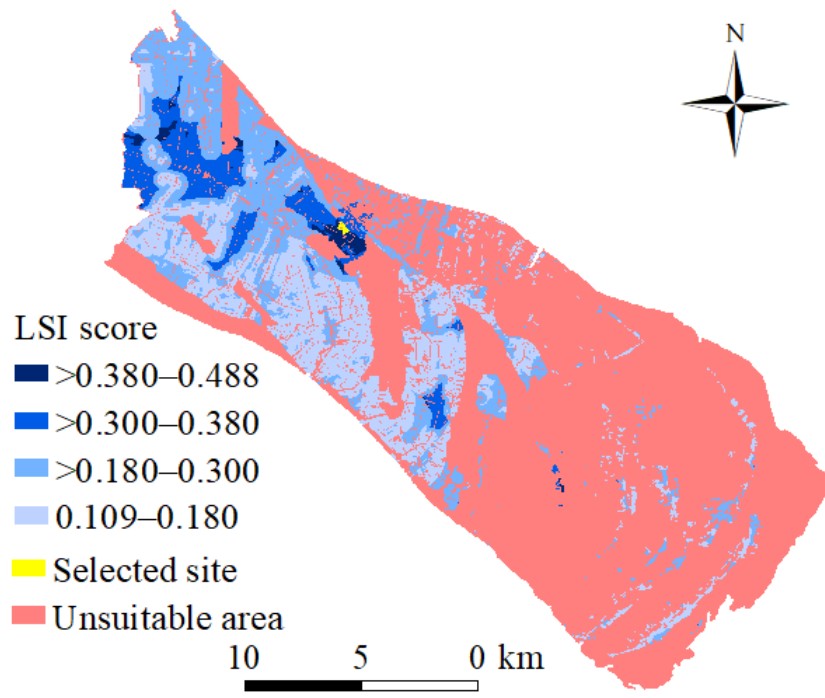

**Figure 5.** LSI score categories and selected site.

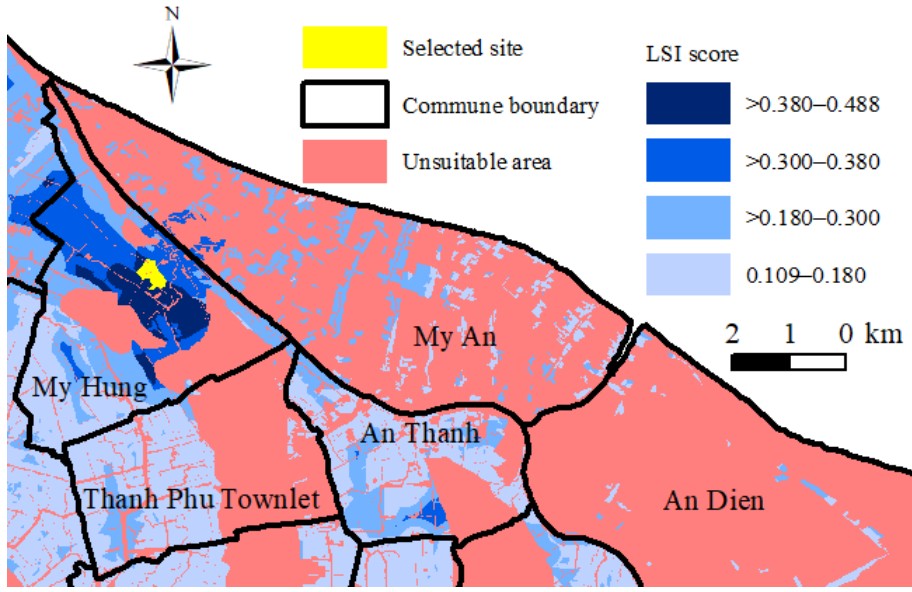

**Figure 6.** Selected site located in My Hung commune.

In addition, the selected site is in the area with a very advantageous geo-environment and geotechnical characteristics which are surface soil of silty clay with few fine sands and over 6 m in thickness as well as the permeability value of $994.49 \times 10^{-9}$ cm/s to $1231.82 \times 10^{-9}$ cm/s. Integrating these three factors: types of soil surface, soil thickness, and permeability as one made the information of soil profile in the study area clear, which is very important in the soil investigation stage. It was the new contribution of this study compared to literature, which only considered a single factor [8,12,20,22,37,44,45,63] or a

couple of factors [15,24,27]. Moreover, the soil thickness and permeability in most previous studies were not presented with certain values.

Except for the unsuitable areas (66.40% of the study area), which indicated prohibited areas based on land use types and sand-containing surface soil layer, the research also created the suitability map based on the LSI score, which showed very advantageous (LSI of >0.380–0.488), advantageous (>0.300–0.380), rather advantageous (>0.180–0.300), and disadvantageous areas (0.109–0.180) in the study area for landfill siting (Figure 7). These areas account for 0.67%, 4.51%, 13.71%, and 14.71% of the study area, respectively.

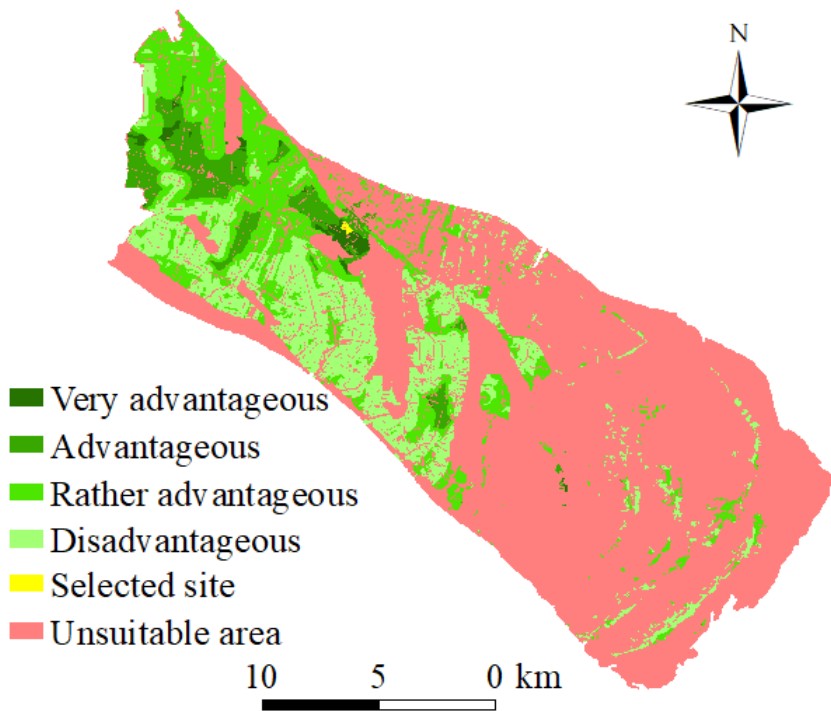

**Figure 7.** Landfill suitability map and selected site in the study area.

## 4. Conclusions

This research proposed a new criterion called geo-environmental and geotechnical characters for landfill siting, which integrated the soil type, thickness of soil surface layer, and soil permeability known as geology related criteria in literature. Considering this criterion made landfill siting more appropriate and effective, especially in the MD where there are Holocene sediments with very soft surface soil. The developed model was run well with eight proposed criteria and pointed out the most suitable site with an area of 14.5 ha, which is expected to serve a population of 100,000–500,000 people and could meet the planned developments in Thanh Phu district in the future. The model may expedite tasks and help to reduce construction cost and time in the first stage of landfill projects, which names investigation and location choice. In addition, the developed model may support local government, policy makers, and managers when proposing suitable planning strategies for landfills, not only in Thanh Phu district but also in other cities or provinces in Vietnam.

In the current research, pairwise comparison, weight calculation and site selection were conducted separately with Excel and ArcGIS software, respectively. Moreover, when the value of pairwise comparison among criteria was changed, it took time to add the calculated weight value to criteria in ArcGIS and ran the model again. Therefore, in further research, the NetLogo software will be applied as a simulation platform in which all works, including pairwise comparison, weight calculation, LSI calculation, and site selection, will be carried out with code and mathematic formulars. Moreover, this time, the research

only focused on the solid waste generated by inhabitants, so other types of waste, such as industrial and hazardous waste, may be considered in future research.

**Author Contributions:** D.-T.N. and M.-H.T. conceptualize the main idea of the paper. T.-P.-U.N., A.-M.L. and Y.Y. prepared the data, tables, and figures. All authors have read and agreed to the published version of the manuscript.

**Funding:** This research is funded by Vietnam National University, Ho Chi Minh City (VNU-HCM) under grant number C2021-18-17.

**Data Availability Statement:** The data presented in this study are available on request from the corresponding author.

**Acknowledgments:** The authors are highly thankful to Vietnam National University, Ho Chi Minh City (VNU-HCM) for financial support. We also acknowledge anonymous reviewers, the assistant editor, managing editor, and the academic editors for their valuable time, productive comments, and suggestions during the review which helped in improving the quality of the manuscript.

**Conflicts of Interest:** No potential conflict of interest was reported by the authors.

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
