# Peer review of "GIS-Based Simulation for Landfill Site Selection in Mekong Delta: A Specific Application in Ben Tre Province"

_remotesensing, doi:10.3390/rs14225704_

Round 1
Reviewer 1 Report (Previous Reviewer 2)
The authors have improved significantly the quality of the manuscript so I suggest accepting it in the current form.
Author Response
Thank you very much for accepting our manuscript
Reviewer 2 Report (New Reviewer)
Although the originality/novelty of the proposed site selection method considering several factors is not so high, I think that there is still valuable and actual contributions to help get a better and more reasonable landfill site selection result or suggestion for the case area.
Author Response
Thank you very much for accepting our manuscript!

Reviewer 3 Report (New Reviewer)
The authors do not hide that research on the location of landfills is prevalent and undertaken by numerous authors. They are well presented in the introduction and the references. The introduction seems a bit long, but it is easy to read, clear and understandable. The authors divided the description of the method into subsections assigned to each analyzed factor. One can feel their commitment to a detailed understanding of the environment in the context of waste disposal. I think that every reader will be able to repeat a given analysis based on the authors' descriptions. The results are presented sparingly. I believe the obtained maps (and the figures generally) need a slight correction. I have doubts about, e.g., the set intervals. I think that the extremes should not overlap. The descriptions of the figures are also very scarce. This is not a mistake, but I suggest the authors expand on them (it will affect the article's visibility after publication). The authors summarized the entire study well.
I have a few questions. Maybe they will help you to slightly improve the discussions and the results.
1) Is the rubbish we are talking about generated by the inhabitants or imported from abroad or other regions?
2) Do you plan to implement your method commercially?
3) I think the relevant offices may be interested in the location designated by you. What is the realistic chance of a landfill being located there?
4) Does the proposed method apply only to communal waste, or can it also be used in industrial and hazardous waste landfills?
I added a few additional comments in the PDF file.

Author Response
Thank you very much for taking the time to read and for your enthusiastic contribution to our manuscript!

Round 2
Reviewer 3 Report (New Reviewer)
Thank you very much for your comprehensive answers. I think the article looks much better now. I believe it can be published in this form. All the best with your project!
This manuscript is a resubmission of an earlier submission. The following is a list of the peer review reports and author responses from that submission.
Round 1
Reviewer 1 Report
The paper is apparently of local interest. From the beginning, the authors wrote about Vietnam and dealt with a particular region in Vietnam throughout the article, except for one section called "Literature review." In the discussion part, the authors only wrote about local things, and no international implication of this work is mentioned.
The insertion of the "Literature review" section does not look appropriate (usually, a brief review is given in the introduction section), and the contents of the review are weak and sometimes strange. For example, the authors cited more than 30 papers in one place by writing "[5-35]," just for AHP in general.
Although too many papers are cited in this particular section, citations in other places are minimal. The most severe problem is the complete lack of citations in the "Results and discussion" and "Conclusions" sections. These sections are where the advantages of this work over the previous work about landfill should be mentioned, and for this purpose, some papers must be cited. But the authors did not provide such an internationally meaningful discussion at all.
The presentation is also problematic. For example, there are 23 figures, which is too many. Also, 20 out of 23 figures are thematic maps for the study area in the same style. Some of such maps should be combined within one layout, and each map will be a subfigure like "Fig. 4A". The authors should learn how to organize a manuscript for an international journal.
Author Response
1/ We have added more the international implication in the result and discussion section as follows
“In addition, the selected site is in the area with very advantageous geo-environment and geotechnical characteristics which are soil surface of silty clay with few fine sands and over 6m in thickness as well as the permeability value of . Integrating these three factors: types of soil surface, soil thickness and permeability as one made clear the information of soil profile in the study area which is very important in stage of soil investigation. It was the new contribution of this study comparing to literature which only considered a single factor [8; 12; 20; 22; 37; 44; 45; 63] or a couple of factors [15; 24; 27]. Moreover, the soil thickness and permeability in most previous studies were not presented with certain values” (Page 14, lines 385-393)
2/ We have deleted the heading “Literature review” and added the contents of “Literature review” section into “Introduction” section. (Page 2-3; lines 66-133)
In this research, we collected 52 papers on landfill site selection and conducted a statistic analysis on methodology, criteria, value categories of each criterion to apply to our research. That was the reason we cited more than 30 papers for “AHP method”. This indicated that the AHP method is the most popular method for site selection.
3/ In the “literature review” section, we reviewed previous studies on used criteria and pointed out the shortage of geology-related criterion. We, then, presented the advantage of this work by proposing a new criterion namely “geo-environmental and geotechnic characteristics” which integrated geology-related criteria including soil types, soil thickness, and soil permeability for landfill siting.
We have added more discussion and more citation in the result and discussion section as follow: “In addition, the selected site is in the area with very advantageous geo-environment and geotechnical characteristics which are soil surface of silty clay with few fine sands and over 6m in thickness as well as the permeability value of . Integrating these three factors: types of soil surface, soil thickness and permeability as one made clear the information of soil profile in the study area which is very important in stage of soil investigation. It was the new contribution of this study comparing to literature which only considered a single factor [8; 12; 20; 22; 37; 44; 45; 63] or a couple of factors [15; 24; 27]. Moreover, the soil thickness and permeability in most previous studies were not presented with certain values” (Page 14, lines 385-393)
4/ We have revised the figures

Reviewer 2 Report
The authors present a methodology for the optimum selection of landfill using GIS and AHP. This methodology is widely used for site selection. Model builder is also a good choice for automating such procedures. However, it is an interesting case-study paper which does not contribute methodologically wise.
The authors mention various AHP methods but choose to implement the “simplest” one but do not argue why. It would be very interesting to implement other variations of AHP and compare the results. This would be a significant contribution.
Also, eight criteria are being selected for the AHP, which are justified sufficiently. Each criterion is presented enough, accompanied by a figure. It does not need such large figures to present these criteria. There is no such detail. On the contrary, there is an extended paper with many large figures, which does not help. It would be better, to decrease the size of these figures and combine them into one large.
In Figure 2, the message is not communicated appropriately. The selected area marked as yellow has the highest LSI score, right? This should be visible on the map. The same stands for unsuitable areas. The whole Figure should be revised.
Last, the reference section contains some weighted lines which seem to be useless. If the authors put them intetionally they should reform properly to communicate the message and align properly all the references.
To conclude, this is a nice case-study paper that I don’t find scientifically original. I suggest accepting the paper after major revision, provided that the authors will work and elaborate on the paper’s novelty.
Author Response
1/ This study did not contribute methodologically wise, but we focused on the new criterion for landfill site selection.
The new contribution of this study was to propose and consider a new criterion namely “geo-environmental and geotechnic characteristics” which integrated geology-related criteria including soil types, soil thickness, and soil permeability for landfill siting. In the literature, the previous studies only considered a single factor of geology criterion, such as soil type/lithology or soil permeability or soil depth/soil thickness or considered a couple of soil type/lithology with soil permeability or with soil depth/soil thickness. Moreover, evaluation of soil permeability and soil depth/thickness was in general. For the soil permeability, most previous authors only based on the soil types to identify the soil permeability, such as clay has low permeability and sand has high permeability. Although they are clays and silts, they were in the different consolidation-levels that made the soil permeability different values, and the thickness also affected the soil permeability. Moreover, the soil depth/soil thickness was not evaluated and presented with certain values. Comparing with the literature, this study based on certain values on soil thickness from boreholes and test results on soil permeability to assess the criterion on geo-environmental and geotechnic characteristics. Considering this criterion made landfill sitting more appropriate and effective especially in the Mekong Delta where is Holocene sediment with very soft surface soil.
2/ We deleted the word “simplest”. The “simplest approach”, here, referred to the way to calculate weight of criteria which were based on pairwise comparisons of the criteria and matrixes of these pairwise comparisons. Moreover, the calculation equations are simple and easy. The sentence, now, is changed as “These verified that the integration of GIS and AHP is the most common approach for identifying suitable sites for landfill in specific and other types of facilities in general, such as, deep-water port [56], solar power farm [57], wind farm [58]” (Page 2, line 79-81).
Your suggestion for comparing results from various AHP methods is a great recommendation. We will try to apply for our future work. However, as mentioned in above response, in this research we only focused on new criterion, namely “geo-environmental and geotechnic characteristics”.
3/ I have revised all the figures indicating the contents of criteria.
4/ Did you mean the figure 22? I have revised the figure 22 by making it bigger in size. It became figure 6
5/ I have deleted the weighted lines in the reference section. This was a mistake when the journal converted my manuscript to the journal’s format.

Reviewer 3 Report
In the present paper the Authors have described the application of a GIS-based simulation for the selection of a landfill site in the Ben Tre Province, by using the analysis hierarchy process (AHP).
The topic is interesting and the manuscript is well-organised. The methodological approach is suitably described. Nevertheless, some clarifications and additional information are needed, as indicated in the following:
1. In the Section 3.3, collected data are presented. Considering the various typologies, formats, and sources of data uses, it would be useful to clarify if have been carried out the pre-processing steps for data homogenisation and normalisation. Such phase, in my opinion, it is fundamental when different data types are used.
2. In the Sub-Section 3.4.5 the calculation of advantage on geo-environmental and geotechnical characteristics (SGGC) is presented. Then, an IDW interpolation has been applied to map the SGGC distribution. I would invite the Authors to justify such choice and, possibly, to indicate if other interpolation methods have been investigated/tested.
3. In the Sub-Section 3.4.8 the criterion about distance from industrial sites is described. The Authors has defined the industrial zones as “places where many people concentrate and work together”. Do such definition include also shopping/business areas (e.g. malls, offices, etc.)? Or is it referred only to factories/plants? In the latter case, I believe that such other zone typologies should be considered within such criterion (unless they are not present: in this case, please, specify this).
4. In the Section 3.5 the analytic hierarchy process (AHP) is described. Considering the judgments used in the pairwise comparison, it would be useful to clarify if they were provided by the Authors or if third-parties Experts have been actually involved. In the latter case, which were their skills? And how were the judgements given and collected (questionnaires, surveys, etc.)?
5. In the “Conclusions” Section it would be important to provide information about the scalability of the developed model in other territorial contexts, and about the future developments (e.g. the implementation of web-platforms and/or Mobile Apps to share the developed model with Experts, Stakeholders, etc.).
6. I would suggest to the Authors to add some further references in the literature review section, concerning the application of WebGIS and AHP in the framework of MCDA (especially when external Experts are involved): DOI 10.1007/978-3-319-42111-7_31, DOI 10.1007/978-3-030-39299-4_38 and DOI 10.14419/ijet.v7i4.15674
7. Finally, a minor remark: the acronym Gis in the title should be indicated in capital letters (GIS, as correctly done in the other parts of the manuscript).
The paper can be accepted after a major review.
Author Response
1/ We have added a sentence to indicate that all data were converted into the same data type of “shapefile” as follows: “Apart from “road network data” which is original “shapefile” data, other data were processed and converted into “shapefile” data before inputting to the developed model” (page 5, lines 179-181)
2/ We have conducted the interpolation with IDW and Kriging methods and found out that the result from IDW interpolation matched with value-known points rather than Kriging interpolation. Therefore, we applied the IDW interpolation for calculation of advantage on geo-environmental and geotechnical characteristics.
3/ In the research, the industrial zones also meant industrial park which included the factories or plants. The shopping/business areas were included within “residential areas” in this research.
4/ The judgment of pairwise comparison was conducted by us based on the literature (page 6, lines 187-188). In this research, we collected 52 papers on landfill site selection and conducted a statistic analysis on methodology, criteria, value categories of each criterion. Based on statistic results, we determined the importance order of the criteria. We, then, defined values from AHP scale for each criterion for the pairwise comparison.
5/ We have added our future research’ information: “In the current research, pairwise comparison, weight calculation and site selection were conducted separately with Excel and ArcGIS software respectively. Moreover, when the value of pairwise comparison among criteria were changed, it took time to add the calculated weight value to criteria in ArcGIS and ran the model again. Therefore, in the further research, the NetLogo software will be applied as a simulation platform in which all works, including pairwise comparison, weight calculation, LSI calculation and site selection, will be carried out with code and mathematic formulars”. (Pages 15-16, Lines 413-419)
6/ We have added references as you recommended as follows “In addition, to enhance public participation for land suitability evaluation, several studies applied the integration of GIS and AHP in the World Wide Web environment which was also known as Web-based multi-criteria spatial decision support system [59, 60] or Web GIS-based multicriteria decision analysis [61]. With these applications, users could select and adjust parameters (criteria) according to their preferences and knowledge which could result in generating different scenarios” (page 2, lines 81 - 87), (page 20, lines 556 - 566)
7/ I have revised the acronym “Gis” into “GIS”. This was a mistake when the journal converted my manuscript to the journal’s format.

Round 2
Reviewer 2 Report
Thanks for your reply. I see improvements in the manuscript but I still have concerns about the contribution. Using an additional layer of information for the AHP and the site selection is not a real contribution. It would be if you could depict the differences in selected sites, some statistics or some descriptive texts could help. Or if you use different methods of AHP and compare the results.
What is more, you describe in your reply your opinion regarding the contribution. What about presenting the contribution in the manuscript? The same question will arise from other readers too. They should be able to find this answer in order to understand why this methodology combined with these criteria is suggested.
In the current form, I still hesitate to accept this manuscript.
Reviewer 3 Report
All my comments provided during the first stage of review have satisfactorily addressed.